# Integrated Nanomaterials and Nanotechnologies in Lateral Flow Tests for Personalized Medicine Applications

**DOI:** 10.3390/nano11092362

**Published:** 2021-09-11

**Authors:** Lucia Napione

**Affiliations:** Department of Applied Science and Technology, Politecnico di Torino, 10129 Torino, Italy; lucia.napione@polito.it

**Keywords:** circulating biomarkers, nanotechnological tools, functionalized surface, device

## Abstract

The goal of personalized medicine is to target the right treatments to the right patients at the right time. Patients with a variety of cancers and other complex diseases are regularly tested as part of patient care, enabling physicians to personalize patient monitoring and treatment. Among the sought-after diagnostic tools, there is an increasing interest and need for those based on a low-cost, easy, rapid, and accurate method for the detection of specific circulating biomarkers above a detection threshold. Lateral flow tests (LFTs), enhanced by nanotechnology, can fulfil these requirements, providing a significant support to personalized patient monitoring. In this review, after a short historical synopsis of membrane-based lateral flow assays, including a description of a typical configuration of a LFT strip, a careful collection is presented of the best characterized nanotechnology approaches previously reported for the enhancement of target detection performance. The attempt is to offer an overview of currently integrated nanotechnologies in LFTs, fostering the actual future development of advantageous diagnostic devices for patient monitoring.

## 1. Introduction

Personalized medicine is a method of clinical practice that takes advantage of new technologies in order to provide decisions concerning the prevention, prediction, diagnosis and treatment of disease [1,2]. Essentially, personalized medicine considers genetic and other biomarker information to make treatment decisions about patients. Disease-related genetic and biomarker information is crucial for a tailored treatment strategy according to which patients receive the most appropriate treatment and routinely undergo biomarker testing as part of their care. This enables physicians to monitor the effect of personalized treatment, improving chances of survival as well as reducing exposure to adverse effects. Therefore, biomarkers can be considered essential tools not only for determining the optimal disease treatment for patients by using the most appropriate drugs, but also for monitoring the patient over time.

In order to guarantee patient monitoring, liquid biopsy—that can broadly be thought as the sampling and analysis of biological fluids for the detection of disease-associated circulating biomarkers—is widely adopted and supported among clinicians, patients and researchers [3,4,5]. At present, the most commonly evaluated biomarkers in liquid biopsy are circulating tumor cells, circulating tumor DNA, exosome and protein biomarkers found in blood samples. For the detection of these biomarkers, several methods and technologies have been developed. In particular, nanotechnology is opening new horizons for highly sensitive and specific detection of circulating biomarkers as well as for drug delivery and clinical imaging.

Nanotechnology can play an essential role in different personalized medicine applications. Indeed, in recent years, an increasing number of nanotechnology strategies have been proposed to overcome limitations in therapeutic drug delivery [6], as well as to improve clinical imaging for disease diagnosis [7], to perform theranostics combining therapeutics and diagnostics [8], or to develop devices for biosensing and/or for interfacing with various downstream biosensors [9,10]. In particular, tremendous effort has been made to develop biosensing systems that exploit nanotechnology to perform the detection of specific circulating biomarkers [11,12]. These biosensors are of special interest for prevention and early disease detection as well as for patient monitoring [13].

Several types of biosensors have been developed to detect biomarkers associated with a variety of diseases [14,15]. Lateral flow biosensors—also known as lateral flow tests (LFTs)—are classified as among the most representative biosensors. LFTs are paper-based devices that can fulfil the requirements expected from a biosensor: specific recognition of the target analyte, robustness, low cost, low amount of sample volume required, quick and user-friendly assay format [16]. Moreover, LFTs can be used for the detection of a variety of biomarkers, such as proteins, nucleic acids and also whole cells. An important aspect of LFT devices is that they enable point-of-care (POC) testing and also at home testing. This may provide a direct benefit for patients and support for personalized medicine. Indeed, current trends in immunodiagnostics are POC tests, and particularly paper-based POC immunoassays, such as LFTs, have attracted great interest in response to the need for widespread qualitative and quantitative biosensing applications. LFTs are well recognized as a mature POC testing technology that has been successfully applied in POC settings [17]. Several LFTs have been reported and commercialized for the detection of various markers in human health diagnosis, as well as in other contexts such as food safety control and environmental monitoring [18,19,20].

Despite their numerous benefits, LFTs have some limitations, in particular concerning their sensitivity. Indeed, sometimes it is hard to obtain the sensitivity required for the detection of a particular target analyte without applying signal amplification strategies. Lateral flow diagnostics are enhanced by nanotechnology [18].

Here, a brief historical synopsis is presented concerning membrane-based lateral flow assays, including a description of a typical configuration of a LFT strip. Then, a carefully selected collection is proposed of the best characterized nanotechnology approaches that have been reported for the performance enhancement of biomarker detection. The main aim of this review is to provide an overview of currently integrated nanotechnologies in LFTs, fostering the actual future development of advantageous diagnostic devices for patient monitoring. Perhaps the most intriguing aspect is that starting from the use of extremely simple materials such as paper (always taken into account for the development of simple detection systems), combing it with the application of nanomaterials and more complex nanotechnologies (only recently developed), and exploiting what is well known from a large part of studies in biochemistry, it is now possible to aspire to the development of promising technologically advanced and at the same time low-cost, rapid, simple/easy-to-use devices that patients with different types of diseases can benefit from.

## 2. Historical Synopsis of Membrane-Based Lateral Flow Assays

Paper is a simple material, yet with great potential. Thanks to its versatility, high abundance and low cost, paper has always drawn great interest as a potential material for sensing in analytical and clinical chemistry [21]. Indeed, science has had a centuries-old interest in using cellulose membranes to perform various measurements in different contexts. Representative examples include: (i) litmus paper (around 1700–1800), a simple piece of cellulose paper impregnated with colored organic compounds obtained from lichens that act as pH sensitive chromophores, enabling the assessment of whether a chemical mixture is acid or alkaline; and (ii) selective spot test paper for metals (around 1930–1940), in which the pH sensitive chromophores are substituted with ligands that are able to change color in the presence of specific metals [22]. Through the years, numerous efforts have laid the foundation for the development of the most widely used form of paper-based assays: the lateral flow assay (LFA).

The development of the LFA—also known as rapid, immunochromatographic test strip—is the result of the convergence of several studies that can be traced back to the 1950s [23]. The technical basis of the LFA was derived from the latex agglutination assay that relies on the formation of cross-linked aggregates as a result of antibody–antigen binding and was first described for the serologic diagnosis of rheumatoid arthritis in 1956 [24]. In the same year, the first paper device for the semi-quantitative detection of glucose in urine was demonstrated [25], and, in the same period, immunoassays were being developed in plate format.

In particular, the development of the LFA was and is essentially linked to the success of different types of immunoassays, all based on the use of antibodies with the ability to specifically bind an antigen that is the target of interest in the assay. Indeed, we can consider the development and implementation of enzyme linked immunosorbent assays (ELISAs) as well as that of the Western blotting (also called immunoblotting) technique as pathfinders for the LFA. Essentially, the LFA takes advantage of the properties of the paper membrane combined with a well-established immuno-detection system and a proper labeling to allow for the rapid detection of the target of interest. The knowledge that nitrocellulose membranes have the capability to bind biomolecules (i.e., proteins and nucleic acids) contributed to the suggestion of allowing liquid samples—containing the analyte of interest, along with antibodies labeled with color particles—to migrate through the small pores of nitrocellulose membranes where other specific antibodies could be adsorbed in a confined area and used for analyte capture and signal detection. This was essentially the idea under the development of the first LFT strips.

The main application that spurred and guided the early development of LFA as a rapid test technology was undoubtedly the human pregnancy test. Starting from notions (i.e., the presence of a specific hormone in the urine of pregnant women, which is human chorionic gonadotropin, hCG) derived from the historical method—based on the injection of the urine of pregnant women into sexually immature rabbits, rats, and mice, which would induce ovarian development—immunoassays were developed and the first commercial lateral flow strip product for a urine-based pregnancy test was launched in the late 1980s. Since then, the application of LFA technology has increased and expanded to a large number of different areas including clinical diagnostics, agri-food, veterinary, industrial testing, environmental health and safety, as well as most recently areas such as molecular diagnostics and theranostics [23].

### Typical Configuration of a LFT Strip

A LFT strip is a paper-based device for LFA, working with the simple principle anticipated above: a liquid sample containing the analyte of interest moves thanks to capillary action (i.e., without the assistance of external forces) through different zones of a polymeric strip, where selected molecules are present and able to interact with the analyte [26]. LFA essentially consists of: (i) a chromatographic system for the separation of components present in a liquid sample on the basis of differences in their movement through a reaction membrane; (ii) conjugation reactions for antibody/probe labeling; and (iii) immune–biochemical reactions for antibody–antigen binding or, more generally, for probe–target analyte binding [27].

A typical LFT strip is represented in Figure 1. It is composed of four main parts: the sample pad, the conjugate pad, the reaction membrane, and the absorbent pad [28]. These components are mounted overlapped on a plastic backing card to ensure better stability and handling. The sample pad (or sample application/deposition pad) is made of cellulose or glass fibers and is located at one end of the strip where the sample to be tested is applied to start the assay. Sometimes, based on the matrix complexity of the sample to be tested, a pretreatment of the sample (before its transportation) is considered, in particular for the separation of different sample components and the removal of interferences. The sample pad should ensure the smooth, continuous and homogenous transportation of the sample and make sure that the analyte present in the sample will be capable of binding to the reagents present in the subsequent pads. The conjugate pad (or conjugate release pad) contains antibodies (or other types of probes) that are specific for target analyte recognition and are conjugated with proper labels/particles, in general colored or fluorescent nanoparticles. The conjugate pad can be made with different materials (e.g., cellulose, glass fibers, polyesters) and should immediately release the label-conjugated probes upon contact with the liquid sample coming from the sample pad, thus allowing the first specific recognition reaction as well as the migration of all the label-conjugated probes towards the following components. The sample, together with the label-conjugated probes free or bound to the target analyte, migrates along the strip into the reaction membrane that is the detection area of the strip. The reaction membrane is usually nitrocellulose, the key material in LFAs. This porous membrane can be functionalized with different biomolecules (e.g., antibodies) that function as capture probes immobilized in lines. More specifically, test and control lines are drawn over the nitrocellulose membrane, corresponding to the first and second line that the sample encounters on its run, and contain probes specific to the target analyte and label-conjugated probe, respectively. The successful detection of the target analyte in the tested sample results in a signal on the test line, while a signal on the control line confirms the proper sample flow through the entire strip and thus validates the performed test. The absorbent pad, located at the end of the strip, contributes to maintaining the flow rate of the liquid sample over the membrane and stops the possible back flow of the sample.

Some of the key points in the design of a LFT strip are: (i) the proper selection of the biochemical reagents to allow specific target recognition; (ii) the choice of the best assay format to perform the LFA (e.g., sandwich format, competitive format, and multiplex detection format); (iii) the proper selection of materials for the different components of the strip; and (iv) the evaluation of possible strategies to enhance signal detection.

Although several different methods have been developed and reported for signal detection, the optical ones represent the most explored thanks to their relative simplicity. In particular, optical methods that do not necessarily require complex equipment to develop and read the signal are of particular interest for the preparation of LFT strips as easy-to-use devices for patient monitoring. However, the simplicity of lateral flow strips designed for easy detection often goes in tandem with poor sensitivity that needs improvement.

## 3. Nanotechnology Approaches for Target Detection Performance Enhancement

The actual performance of a diagnostic test relies in particular on its specificity and sensitivity. In general, LFT devices have low performance in comparison to gold standard techniques, such as ELISA and reverse transcription polymerase chain reaction (RT-PCR). Figure 2 shows schematically the main key items to be considered for LFT improvement in terms of both specificity and sensitivity.

High specificity of LFTs is mainly ensured by (i) selection and screening of the biochemical reagents devoted to specific target recognition (i.e., highly specific affinity molecules, such as antibodies, receptor proteins, aptamers, nanobodies, short peptides) and (ii) by assay optimization to minimize nonspecific binding events (e.g., removal/reduction of substances capable of nonspecific binding, surface functionalization/blocking of labels, and nitrocellulose membrane blocking) [29] (Figure 2, upper portion).

Nanotechnology does not currently have a direct and significant impact on LFT specificity compared to the decisive one it has on LFT sensitivity. However, the use of nanobodies can be mentioned here—more properly as nanobiotechnological tools—contributing to the improvement of LFT specificity. Nanobodies are single domain antibodies derived from camelid atypical immunoglobulins and presenting unique properties including nanoscale size, high specificity and affinity for only one cognate target, lower cross reaction and ability to recognize difficult-to-access antigen epitopes [30]. Examples of previously reported works on LFTs employing nanobodies are the development of: (i) a nanobody-based LFA for detection of human norovirus [31]; (ii) a nanobody-based LFA to detect active *Trypanosoma congolense* infections with high specificity [32]; and (iii) a serum-stable, nanoparticle catalyst-labeled LFA with proper modified nanobodies having high affinity and specificity toward p24 antigen, the viral capsid protein of HIV [33]. Of note, as reported by Loynachan and colleagues (i.e., the authors of the abovementioned work), the use of nanobodies can be combined with that of a nanotechnology signal amplification strategies to successfully obtain a device characterized by high specificity and sensitivity for target detection.

High sensitivity of LFTs can be achieved considering in particular (i) assay improvement by optimization of assay kinetics (which in practice in most cases consists of increasing reactant concentrations directly or indirectly and in increasing the reaction time) and signal amplification strategies (including physical/chemical enhancement and reader use), and (ii) sample enrichment [29] (Figure 2, bottom portion).

As anticipated above, nanotechnology has a decisive impact on LFT sensitivity, and this can mainly be attributed to the development of different nanotechnology approaches for signal amplification leading to target detection performance enhancement [34,35,36,37]. These approaches are essentially based on the design/selection of different labels that generate the signal, detectable by the naked eye or through a proper reader. Some label designs preserve all the well-known advantages of traditional LFTs, others reduce in part these benefits.

At present, gold nanoparticles (GNPs) are the most used labels in commercially available LFA-based devices. This is essentially due to the (i) ease of synthesis and conjugation with molecular probes; (ii) high chemically stability; and (iii) optical properties, allowing the generation of an intense color detectable by the naked eye [38]. However, GNP colorimetric signal is limited by localized surface plasmon resonance (LSPR) activity and a lack of amplification [39]. This means that a defined amount of GNP produces a fixed amount of signal that may not always be sufficient to obtain the desired sensitivity for detecting the target of interest. To improve LFT sensitivity, different signal amplification strategies have been developed. Among them, the best characterized and simplest nanotechnology approaches include the dual gold signal enhancement method, the use of carbon nanotubes (CNTs), carbon nanoparticles (CNPs) and nanozymes. These approaches are essentially based on proper labeling strategies taking advantage of the properties of different types of nanomaterials. In general, these methods do not require either complex or expensive equipment for signal detection and are implemented for colorimetric readouts, suitable for naked eye detection as well as for detection using a simple LFT reader. In the following, the description of the abovementioned nanotechnology approaches is provided along with representative examples of previously reported works. In addition, considering the possibility of using different sensing modalities, the last part of this section deals with label designs combined with the use of proper—in general more complex—reader systems, enabling increased sensitivity.

### 3.1. Dual Gold Signal Enhancement Method

Two populations of GNPs can be used to achieve signal enhancement according to a double gold conjugation technique applied to LFTs (see Figure 3). The first GNP population is conjugated with a molecular probe specific for analyte recognition, while the second one is designed to bind only with the first GNP population. In general, these two GNP populations are of different size and positioned at different sites (i.e., first and second conjugate pads) to avoid mixing before the assay. The binding between the two GNP populations can be achieved using specific antigen–antibody reactions, primary and secondary antibody reactions, or other binding systems, such as those based on exploiting biotin–streptavidin binding affinity. The signal is enhanced due to twice the aggregation of GNP conjugates. Two of the most representative studies are reported below Figure 3.

Choi and colleagues developed a dual GNP conjugate-based LFA method for the analysis of troponin I, a biomarker of myocardial injury [40]. The first GNP population was conjugated with an antibody specific for troponin I recognition and blocked with bovine serum albumin (BSA), while the second GNP population was conjugated with anti-BSA antibody, allowing the binding with the BSA blocked nanoparticles. In particular, the authors highlighted that the size of the two GNP populations was crucial for the sensitivity improvement. More specifically, they found that the use of 10 nm and 40 nm size (for the first and second GNP population, respectively) was able to ensure a detection sensitivity increase of about a 100-fold compared to the conventional LFA. The developed test detected as low as 0.01 ng/mL troponin I in a few minutes and was successfully applied in the analysis of biological samples (i.e., blood serum) from patients with myocardial infarction.

Shen and colleagues developed a LFA with dual GNP conjugates for the detection of hepatitis B surface antigen (HBsAg), a biomarker of virus replication in serum [41]. The first GNP population was conjugated with biotin and an antibody specific for HBsAg recognition, while the second GNP population was streptavidin-conjugated. The assay was used to detect HBsAg in the range from 0.1 to 30 ng/mL and the detection sensitivity was found to be increased about 30-fold compared to conventional LFAs.

In general, the main advantage of the dual gold signal enhancement method is that it ensures good sensitivity improvement without the need for additional operation steps, thus preserving the LFT benefits of being easy to use, rapid, and low-cost.

### 3.2. CNTs and CNPs

The use of GNPs can be replaced in LFTs by that of other nanomaterials having stronger colorimetric contrast, such as carbon nanomaterials (see Figure 4) comprising different types of nanostructured products based on carbon black. The intrinsic black color of carbon nanomaterials ensures the highest possible visual contrast in the nitrocellulose membrane, thus providing improvement of the LFT sensitivity [42,43,44]. In particular, CNTs and CNPs have gained significant attention as labels for LFTs [45]. Besides their optical features, CNTs and CNPs are low-cost materials and have the appropriate reactivity and stability required for LFT labels. Two representative studies are reported below Figure 4.

Qiu and colleagues reported on a CNT-based LFA for the ultrasensitive detection of proteins using rabbit immunoglobulin G (IgG) as a model target in order to demonstrate the proof-of-concept [46]. More specifically, the working principle of the developed LFT was based on the use of (i) rabbit IgG as target analyte, (ii) goat anti-rabbit IgG as both detection probe conjugated with CNTs and capture antibody immobilized on the test line (exploiting the capability of polyclonal antibodies to interact with more than one site), and (iii) donkey anti-goat IgG as capture antibody immobilized on the control line. This system resulted in a detection limit of 1.3 pg/mL, over three orders of magnitude lower than that of previously reported GNP-based LFTs. Moreover, the authors successfully demonstrated the clinical diagnostic feasibility of the developed CNT-based LFT by detecting the analyte in spiked blood plasma samples.

Porras and colleagues reported on a comparative study of GNPs and CNPs in nucleic acid LFA [44]. They presented the conjugation of avidin to CNPs (Av-CNPs) and the design of LFAs for the detection of PCR products in order to compare the performance of Av-CNPs to commercial streptavidin-conjugated GNPs (streptAv-GNPs). More specifically, (i) biotin and digoxigenin were used as tags to label the PCR-amplified DNA, (ii) CNPs were conjugated with avidin and compared with streptAv-GNPs, (iii) an anti-digoxigenin antibody was immobilized on the test line, and (iv) a biotynilated protein was deposited on the control line. The results of the comparative study indicated an improved limit of detection for Av-CNPs compared to streptAv-GNP, attributed to the higher signal background contrasts thanks to the intensity of the black color.

In general, as in the case of the dual gold signal enhancement method, the use CNTs and CNPs maintains the above-mentioned LFT benefits.

### 3.3. Nanozymes

The use of enzymes has been proposed as a chemical enhancement for signal amplification in LFTs [47]. Usually, catalytic amplification is obtained by exploiting natural enzymes such as horseradish peroxidase (HRP) to catalyze oxidation–reduction reactions at the level of the test and control lines [48,49,50]. However, natural enzymes have some limitations that are essentially related to their sensitivity to the environment (e.g., temperature, pH, possible presence of interfering molecules/agents in the biological matrices). Recently, the use of nanozymes has been proposed as an alternative to natural enzymes in order to overcome these limitations. Nanozymes are defined as nanomaterials with enzyme-like properties, offering higher catalytic stability, ease of modification and lower cost than natural enzymes [51]. They can catalyze different enzymatic reactions mimicking the activities of natural oxidases, peroxidases and catalases [52]. In particular, nanozymes with intrinsic peroxidase-like activity, e.g., platinum (Pt) nanoparticles, have gained significant attention thanks to their possible integration in biosensors, including LFTs (Figure 5). Three representative studies are reported below Figure 5.

Gao and colleagues reported on Pt-decorated GNPs with dual functionalities for ultrasensitive colorimetric in vitro diagnostics (IVD) [53]. GNPs were engineered by coating with ultrathin Pt skins of sub-10 atomic layers resulting in a product (i.e., Au@Pt NPs) that was demonstrated to retain the plasmonic activity of initial GNPs while possessing ultrahigh catalytic activity enabled by Pt skins. Appling this system to LFT as a model of IVD, the authors proposed two different detection alternatives: a low-sensitivity mode produced just by the color from plasmonics and a high-sensitivity mode (i.e., sensitivity enhancement of two order of magnitude) ensured by the color obtained from the catalytical activity of the nanozyme in the presence of a chromogenic substrate, thus offering an “on-demand” tuning of the signal detection performance. Moreover, the authors successfully demonstrated the potential clinical use of the developed Au@Pt NP-based LFT, applying it to quantifying human prostate-specific antigen (PSA)—a well-known cancer biomarker—from human plasma samples.

In a similar study, Zhang and colleagues described the preparation of a gold–Pt nanoflower (AuPt NFs) showing that they can be used both as a label and as an artificial enzyme in LFTs [54]. The authors prepared AuPt NFs by growing Pt nanowires on the surface of GNPs and used rabbit IgG as a model analyte (this latter as in the work previously reported [46]). The found detection limit was 100 times lower than that of the conventional GNP-based LFTs.

It is recalled here—further describing it—one of the above-mentioned examples of previously reported works on LFTs employing nanobodies, that is the study performed by Loynachan and colleagues [33]. The authors reported on the synthesis and characterization of porous Pt core–shell nanocatalysts (PtNCs) that showed high catalytic activity and explored the application of nanobody-functionalized PtNCs having high affinity and specificity toward p24, establishing the key larger nanoparticle size regimes needed for efficient signal amplification and performance in LFT. The catalytic amplification due to the use of PtNCs enabled the naked-eye detection of analyte spiked into sera in the low femtomolar range (ca. 0.8 pg/mL), thus surpassing the published sensitivity of the available commercial rapid tests for p24.

In terms of maintaining of the well-known LFT benefits, it is to be considered that in general when using nanozymes (as in the case of natural enzymes) an additional operation step, i.e., the delivery of the chromogenic substrate, is required in comparison to traditional LFTs. However, ad hoc solutions can be developed, including for instance the pre-deposition of the substrate in the nitrocellulose membrane or its dispensing through preloaded cartridges [47]. In addition, besides the colorimetric technique, other detection methods can be used (e.g., chemiluminescence) depending on the selected nanozyme/substrate system.

### 3.4. Label Design Combined with Reader Use

Nanomaterial labels in LFT can be designed not only for colorimetric readout, but also for producing a different kind of signal, usually requiring the use of a proper reader and enabling increased sensitivity (see Figure 6). In general, after the assay, labels are excited by a physical stimulus and then signal is acquired using a proper detector. The detection methods for high-sensitivity LFTs may be of different types (e.g., optical, magnetic, and thermal) and are classified based on the different sensing modalities exploited, according to the selected labels and recognition/amplification strategies [55].

Optical methods are the most widely developed methods for LFTs. Among them, fluorescence, surface enhanced Raman scattering (SERS), and luminescence have been considered as sensing modalities besides the colorimetric technique.

LFTs based on novel fluorescent labels such as quantum dots (QDs) have been receiving increasing attention. QDs are semiconductor nanocrystals, exhibiting unique optical and electronic properties that are different from bulk material due to quantum mechanics [56]. Compared with conventional dye fluorescent labelling materials, QDs have narrow and symmetric photoluminescence spectra, strong fluorescence emission intensity, high quantum yields, and robust photostability. As a representative example, the work reported by Chen and colleagues [57] is mentioned. They developed a novel QD-doped polystyrene nanoparticle-based LFT to detect two cancer biomarkers, i.e., cytokeratin-19 fragment (CYFRA 21-1) and carcinoembryonic antigen (CEA) in human serum samples. The authors prepared the QD-doped carboxylate-functionalized polystyrene nanoparticles used as fluorescent reporters in the LFT system that was based on a sandwich immunoassay and associated with a fluorescence strip reader. The results indicated that the developed QD-based LFT system is characterized by good stability and high sensitivity in comparison with conventional methods.

Functional SERS-encoded nanoparticles (also called SERS nanotags) can be integrated into LFTs [58,59,60]. SERS—a surface-sensitive detection technique based on the enhancement of Raman scattering by molecules adsorbed on nanostructured metal surfaces—is undoubtedly one of the most sensitive detection methods for LFT. However, the main limitations are the time taken to scan the signal detection area and the need for expensive equipment, including a dedicated Raman laser and a high-performance spectrometer [61]. As a representative example, the work reported by Gao and colleagues [62] is mentioned. They developed a SERS LFT for the detection of neuron-specific enolase (NSE), a biomarker of brain injury (and also a biomarker for tumors such as neuroblastoma and small cell lung cancer) in blood plasma. More specifically, the authors used SERS probes in which the Raman reporter molecules are sandwiched between an Au nanostar and a thin silica shell. The developed SERS LFT exhibited superior performance in terms of sensitivity compared with colorimetric LFTs.

With regard to luminescence detection for LFTs—besides the chemiluminescence methods based on the use of enzymes and proper substrates—the use of persistent luminescence nanoparticles (PLNPs) has been reported, which are unique optical nanomaterials emitting afterglow luminescence after ceasing excitation [63]. In particular, Paterson and colleagues demonstrated that europium- and dysprosium-doped strontium aluminate PLNPs ensure superior analytical sensitivity than GNP in LFTs [64]. More specifically, they considered NeutrAvidin PLNPs as a labelled probe, biotinylated lysozyme as a model analyte in buffer, and anti-lysozyme antibodies immobilized on the test line as capture antibodies. Using a standard gel reader for luminescence imaging, the authors achieved a limit of detection approximately one order of magnitude more sensitive than GNP. Moreover, the same research group reported on a low-cost smartphone-based platform for LFTs with persistent luminescent nanophosphors [65]. In this latter work, the authors used hCG as a target model and proposed the use of the phone’s flash for excitation of the nanophosphors and the phone’s camera for prompt, time-gated imaging of the persistent luminescence. This proposed imaging strategy resulted in detection limits in line with other photoluminescent labels, but with remarkably simpler and less expensive hardware for the readout.

Besides those for optical methods, nanomaterial labels and readers for magnetic and thermal detection have also been reported.

Magnetic nanoparticles (MNP) can be used as detecting labels that can be sensed by means of different external devices (magnetoresistive readers and inductive readers) and at the same time as agents for magnetic separation devoted to analyte preconcentration prior to LFA [66]. As an example, the work by Lu and colleagues [67] is mentioned. These authors reported on a magnetic nanobeads-based LFT developed for the simultaneous detection of two biomarkers, i.e., NSE and CEA, sensitive in the clinical diagnosis of small cell lung cancer. As a result, the calculated limit of detection was 0.094 ng/mL and 0.045 ng/mL for NSE and CEA, respectively, (i.e., sensitivity enhancement of one order of magnitude compared to naked eye detection). Moreover, the authors used one hundred and thirty clinical samples to validate the LFT system that exhibited high specificity and sensitivity.

Metallic nanoparticles (MeNPs) can also be used as detecting labels by exploiting their thermal response under external stimulation. In general, a NIR laser source and an infrared camera are required. As an example, the work of Hu and colleagues [68] is mentioned. They developed a calorimetric LFT based on the use of gold nanocages, selected in particular for their high photothermal conversion efficiency and stronger penetrating ability of NIR light. The authors established and validated the proposed calorimetric LFT using two target analytes, alpha-fetoprotein, the most widely used serum marker for liver cancer, and zearalenone, a food toxin. The proposed calorimetric LFT showed superior performance when compared with conventional visual LFTs.

## 4. Conclusions

The presented careful collection of the best characterized nanotechnologies integrated in LFTs shows how nanomaterial-based approaches can actually lead to a significant enhancement in LFT target detection performance.

In the context of personalized medicine applications—for the prevention, prediction, diagnosis and treatment of cancer and other complexes diseases—an ideal LFT should have all the well-known benefits of the traditional LFT in addition to substantially improved sensitivity. In other words, an ideal LFT should be low-cost, easy, rapid, accurate and highly sensitive. For these motivations, among the presented nanotechnology approaches, those that do not require either complex or expensive equipment for signal detection are the most promising to be fostered for effective and widespread employment in the near future in the context of personalized medicine applications.

In addition, an important consideration when dealing with diagnostic devices for personalized medicine applications is the possibility to actually quantify the readout of the test. Therefore, integrated nanotechnologies in LFTs should allow easy signal quantification, semi-quantitative or fully-quantitative according to the type of biomarker considered.

In conclusion, recalling and summarizing the words mentioned at the end of the introduction section, the most intriguing aspect is the successful combination of a simple paper-based system with the use of advanced but easy and low-cost nanomaterial-based approaches leading to the development of promising advanced LFT devices to be fostered for personalized medicine applications.

## Figures and Tables

**Figure 1 nanomaterials-11-02362-f001:**
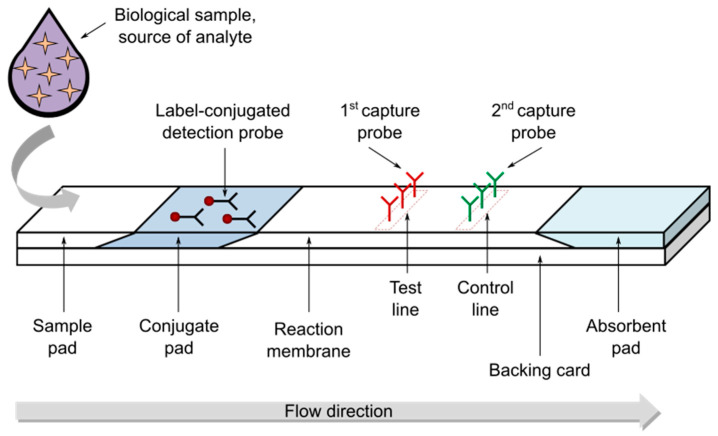
Illustration of a typical lateral flow test (LFT) strip.

**Figure 2 nanomaterials-11-02362-f002:**
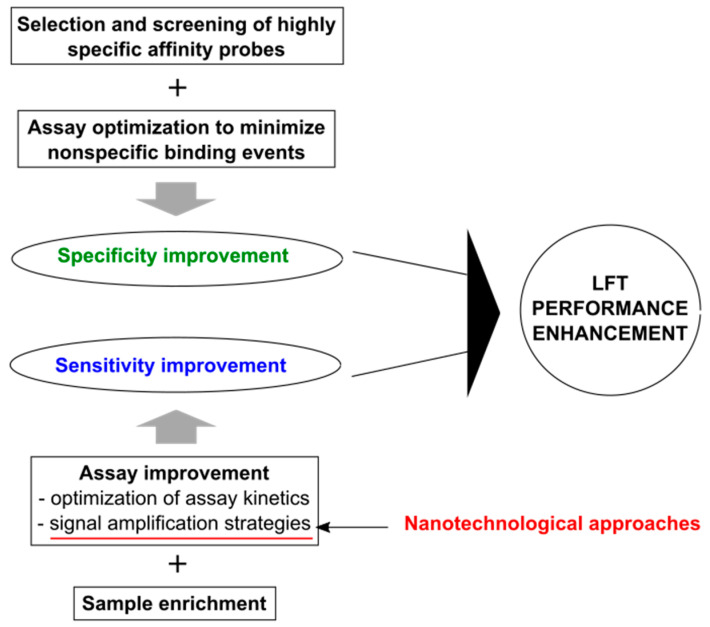
Schematic of the main points under the enhancement of LFT detection performance. The main contribution of nanotechnological approaches is highlighted in red.

**Figure 3 nanomaterials-11-02362-f003:**
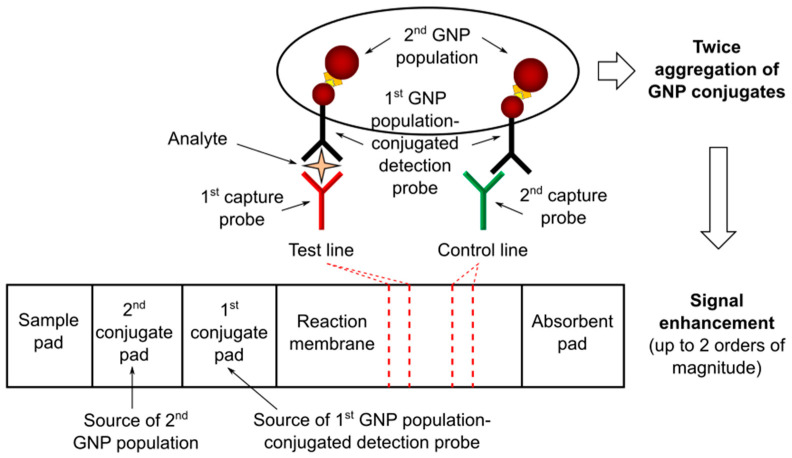
Schematics of dual gold signal enhancement method applied to LFTs. GNP, gold nanoparticle.

**Figure 4 nanomaterials-11-02362-f004:**
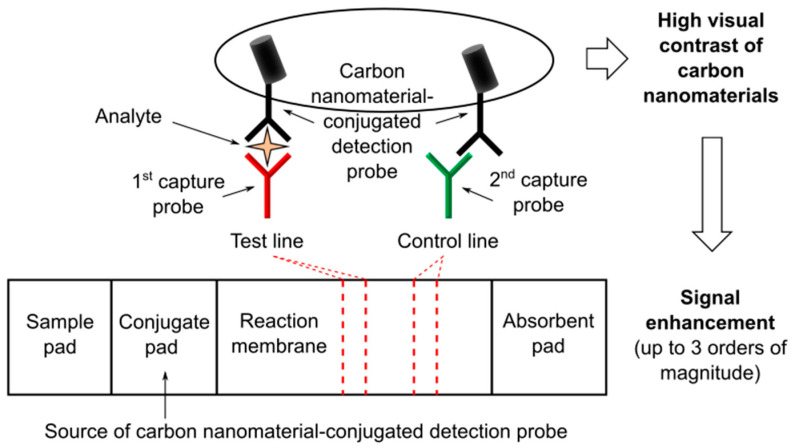
Schematics of a LFT based on the use of carbon nanomaterials, e.g., carbon nanotubes (CNTs), as labels for signal enhancement.

**Figure 5 nanomaterials-11-02362-f005:**
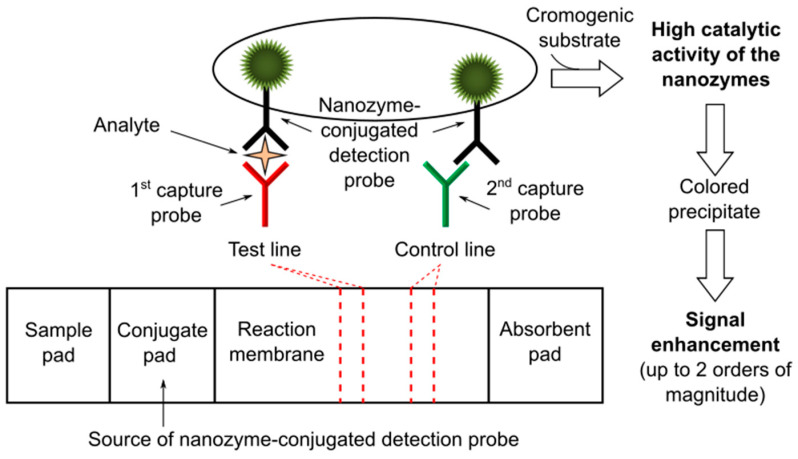
Schematics of a LFT based on the use of nanozymes, e.g., platinum (Pt) nanoparticles, as labels for signal enhancement.

**Figure 6 nanomaterials-11-02362-f006:**
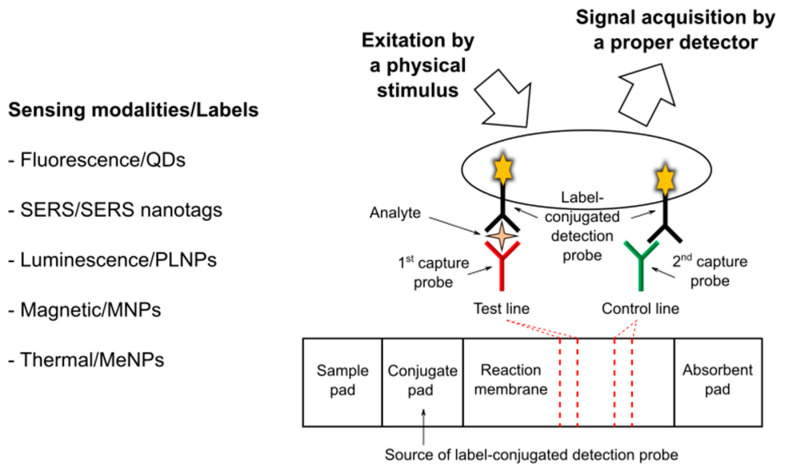
Simplified schematics summarizing the main sensing modalities and associated labels for high-sensitivity non-colorimetric detection methods applied to LFTs. QDs, quantum dots; SERS, surface enhanced Raman scattering; PLNPs, persistent luminescence nanoparticles; MNP, magnetic nanoparticles; MeNPs, metallic nanoparticles.

## Data Availability

The study did not report any data.

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
