# Peer review of "Integrated Nanomaterials and Nanotechnologies in Lateral Flow Tests for Personalized Medicine Applications"

_nanomaterials, 2021, doi:10.3390/nano11092362_

Round 1

Reviewer 1 Report

The author reviewed the popular Integrated Nanotechnologies in Lateral Flow Tests for Personalized Medicine Applications. The scientific content of this review paper is substantial and arranged in a good logic flow. There are only three suggestions here. First, all the sentences expressed with first person should be rephrased to passive voice. Second, the title can be further modified to strengthen the importance of nanomaterials, e.g. “Integrated Nanomaterials and Nanotechnologies in Lateral Flow Tests for Personalized Medicine Applications”. Third, some recent literatures on nanotechnology can be properly cited in this paper, e.g. [https://doi.org/10.1002/elps.202000110]. This paper can be accepted for immediate publication in this renowned international journal after minor revisions.

Reviewer 2 Report

As one of the most representative biosensors, lateral flow tests have the advantages of low cost, good robustness, small sample volume, strong practicability and user friendliness. Although LFTs can be used for the detection of a variety of biomarkers, such as proteins, nucleic acids and also whole cells, LSTs have still some challenges, in particular concerning their sensitivity. This review presents a brief historical synopsis concerning membrane-based lateral flow assays, and provides an overview of currently integrated nanotechnologies in LFTs for the enhancement of target detection performance. Overall, this manuscript is well written and it can be published after minor revision. In part 3, the content described in subtitle 3.4 is not quite consistent with the object to be described in part 3, and it intersects with the contents of section 3.1 to 3.3. This makes the logic of this part a little confused, and it is difficult for readers to understand the focus of this part.

Reviewer 3 Report

The author described different nanotechnologic strategies applied to Lateral Flow Tests.

Although the manuscript is easy to follow, it vaguely describes some important enhacement methods without discussing them in depth, which it is an unfortunate since the reader is expecting to read them in detail due to the scope of the journal. It also lacks of some recent reviews (published in other journals) about nanotechnologies applied into LFT.

Unfortunately, this work does not contribute significantly to the field. This work, in its current form, it is more appropiate for journals focused on biosensors.
